

# Kinesiophobia among health professionals' interventions: a scoping review

Lilian Doutre[1,2], Maryse Beaumier[2,3], Andrée-Anne Parent[2], Sébastien Talbot[2] and Mathieu Tremblay[2]

[1] Programme d'activité physique adaptée et santé, Université Rennes 2, Rennes, France
[2] Department of Health Sciences, Université du Québec à Rimouski, Rimouski, Quebec, Canada
[3] Research Center of Centre intégré de santé and services sociaux de Chaudière-Appalaches (CISSS-CA), Québec, Québec, Canada

## ABSTRACT

Health professionals are regularly confronted with patients suffering from a fear of movement-related pain (unknown as kinesiophobia). The fear-avoidance attitudes and beliefs of healthcare professionals are likely to play a key role in their patients' therapeutic approach. However, kinesiophobia among health professionals is a relatively young topic. This scoping review aims to explore and catalogue the extent of scientific research that identifies the causes and consequences of kinesiophobia among health professionals while they perform their interventions. The review was based on the Joanna Briggs Institute manual and the PRISMA method for a scoping review. The research was conducted in May 2024 using CINHAL, Medline and Sportdiscus databases with the search terms "fear-avoidance", "kinesiophobia", "pain-related" and "physical therapist". Out of 2,162 potential studies, thirteen articles were included. No study directly mentioned kinesiophobia among health professionals, but it was studied through fear-avoidance beliefs. Two-thirds of the articles indicate that professionals with fear-avoidance beliefs tend to refer their patients to other specialists less frequently and limit their patients' activity, despite treatment guidelines. Most of the studies found were physiotherapists' interventions for chronic back pain patients. The current review emphasizes the need for additional studies involving more healthcare professionals and diverse health conditions.

Corresponding author
Mathieu Tremblay,
mathieu_tremblay2@uqar.ca

## INTRODUCTION

Kinesiophobia is defined as "*an excessive, irrational and debilitating fear of movement and activity resulting from a sense of vulnerability to painful injury or re-injury*" (*BaykalŞahin, Kalaycíoğlu & Şahin, 2021*). In the literature, this avoidance behaviour caused by threatening motion experience (*Bak et al., 2022*) is observed in a wide range of chronic conditions (such as musculoskeletal pain, osteoarthritis, fibromyalgia, cardiorespiratory pathologies) among the general population (*BaykalŞahin, Kalaycíoğlu & Şahin, 2021*; *Acar, Sönmezer & Yosmaoğlu, 2022*; *Perles et al., 2022*; *Shen et al., 2023*), but also among injured

athletes (*Hsu et al., 2017*). Increased symptoms of kinesiophobia are associated with a decreased likelihood of returning to sports after surgery (*Kvist et al., 2005*; *George et al., 2012*; *Vascellari et al., 2021*). Over time, an individual who suffers from kinesiophobia symptoms can experience social isolation, physical disability, as well as depression and anxiety symptoms (*BaykalŞahin, Kalaycíoğlu & Şahin, 2021*; *Shen et al., 2023*).

The fear-avoidance model (*Conradi & Masselin-Dubois, 2019*) helps to explain the link between pain-related fear and safety-seeking behaviours amongst the patients (*Bisson et al., 2022*; *Siddall et al., 2022*; *Xu et al., 2020*). The model states that if the pain caused by exercise or movement is interpreted as threatening (catastrophizing), then pain-related fear progresses, leading to avoidance behaviour that results in physical disability, which is responsible for the persistence of the vicious cycle of kinesiophobia (*Bisson et al., 2022*; *Siddall et al., 2022*; *Xu et al., 2020*). Health professionals are on the front line to help their patients to break that vicious cycle of kinesiophobia. To our knowledge, only a limited number of studies have investigated the fear-avoidance model and the impact of health professionals. Therefore, a better understanding is still required to help these professionals during their interventions.

Kinesiophobia observed among patients is a well-known phenomenon that continues to be studied as a measure of treatment efficiency (*Lanhers et al., 2020*). In contrast, the study of kinesiophobia among health professionals and its influence on their patients is a relatively young topic (*Lakke et al., 2015*). In fact, the recognition of the importance of the relationship between the professional and patient dates before Sigmund Freud (1856–1939) (*Ardito & Rabellino, 2011*). Nowadays, the therapeutic alliance model, also called the therapeutic relationship model, highlights the importance of this relationship in a successful intervention plan (*Baillargeon, Pinsof & Leduc, 2005*). The model's concept is based on the relationship between professional and patient, where both sides' feelings, thoughts and response predispositions are bound together (*Pinsof, 1995*). By following the therapeutic alliance model, health professionals are most likely to have positive influences on their patient's intervention. This positive influence on patients is observed across several outcomes by improving the ability to perform daily activities, specific functional tasks, overall physical health and overall satisfaction with the treatment, and also by reducing pain and depression symptoms (*Hall et al., 2010*). Through the therapeutic alliance model, it is understood that the health professionals are directly involved in the success of the intervention plan. However, the attitudes and beliefs of professionals are likely to play a key role in the treatment approach of their patients (*Gardner et al., 2017*; *Lakke et al., 2015*; *Bell, 2022*). Indeed, professionals with kinesiophobic behaviours might transfer their fear and belief to their patients through verbal and non-verbal behaviours (*Blouin et al., 2019*; *Farris et al., 2019*; *Fujii et al., 2019*).

Given the current limited understanding of this topic, our study aims to fill a crucial gap by comprehensively mapping the existing scientific research on kinesiophobia among health professionals and their impacts on intervention efficacy. The term ''intervention'' in current research pertains to therapeutic practices involving physical activity or exercise routines prescribed by health professionals to address health concerns. To this date, no scoping reviews have been found on this topic. This review not only identifies the

causes and consequences of kinesiophobia but also delves into kinesiophobic behaviour or fear-avoidance behaviour exhibited by health professionals during interventions. Notably, our investigation focuses on the professional's influence on the patient rather than solely on patient outcomes, thus providing a novel perspective. This comprehensive review serves as an indispensable resource for health professionals, students, scholars, and scientists eager to expand their understanding of kinesiophobia among healthcare providers and its detrimental effects on interventions. Importantly, our scoping review represents the first of its kind on this topic, highlighting its novelty and significance in the field.

## METHODS

The methodological framework followed the steps of the Joanna Briggs Institute (JBI) Manual for Evidence Synthesis (*Aromataris & Munn, 2020*) and the Preferred Reporting Items of Systematic Reviews and MetaAnalyses—Extension for Scoping Reviews (PRISMA-ScR) for transparent research (*Tricco et al., 2018*).

### Identifying the research questions

Two key questions were formulated for this scoping review. The first question was: What were the potential reasons for health professionals experiencing kinesiophobia or displaying kinesiophobic behaviour? By investigating the potential reasons health professionals experience kinesiophobia or display kinesiophobic behaviour, we aim to identify the underlying factors contributing to this phenomenon. Understanding these reasons is essential for developing targeted interventions and support systems to mitigate kinesiophobia among healthcare providers, thereby improving the quality of care and patient outcomes. The second question was: What were the possible outcomes on the patient attitude of health professionals' experiencing kinesiophobia or displaying kinesiophobic behaviour during their interventions? This research question focuses on the possible outcomes on patient attitudes resulting from health professionals experiencing kinesiophobia or displaying kinesiophobic behaviour during interventions. This inquiry is crucial for comprehensively assessing the impact of kinesiophobia on patient-provider interactions and healthcare outcomes. By elucidating the consequences of kinesiophobia from the patient's perspective, we can better understand the broader implications of this phenomenon on patient care and satisfaction. Within the broader context of the field, these questions can highlight their implications for clinical practice, education, and policy development during health interventions.

### Identifying relevant studies

The establishment of the study criteria was broad to target the topic of the study appropriately using Mnemonic PCC (Population, Concept, Context) as a framework (*Aromataris & Munn, 2020*). Population : Health professionals suffering from kinesiophobia's characteristics; Concept : To determine whether health professionals suffer from kinesiophobia's characteristics based on factors such as fear-avoidance, pain, physical disability, deconditioning, fear of pain, self-efficacy or catastrophizing that create kinesiophobic behaviours and/or beliefs and/or knowledge that impact their

interventions (*Baykal Şahin, Kalaycíoğlu & Şahin, 2021*; *Ding et al., 2022*; *Sharpe et al., 2022*); Context : Within the intervention of health professionals.

## Studies selection

The scoping review has been conducted on the CINHAL, Medline, and Sportdiscus databases as of May 2024. An experienced health sciences librarian helped conduct searches on the databases. The librarian identified the most relevant databases based on the research questions and the review topic. Searches were limited to French and English language citations. We combined the following keywords using Boolean operators to ensure no relevant studies were missed. Appendix S1 details the Boolean operators for each database. The following keywords were used: ''Pain-Related Fear'', ''Avoidance behaviour'', ''Pain belief'', ''Fear of movement'', ''Kinesiophobia'', ''Fear avoidance'', ''Anticipation of pain'', ''Fear'', ''Movement'', ''Occupational therapists'', ''Nurses'', ''Physical Therapists'', ''Physicians'', ''Physicians, Family'', ''Rheumatologists'', ''Physical therapist'', ''Physiotherapist'', ''Kinesiotherapist'', ''Clinician'', ''Health professional'', ''Health personnel'', ''Practitioner'', ''Chiropractor'', ''Physician'', ''Health care provider'', ''Rheumatologist''. Thus, reference lists of articles identified in the databases were reviewed.

## Data extracting and mapping

A data charting form in Microsoft Excel was utilized during the data extraction process. This form was provided to the evaluators with a descriptive summary of the results (Table 1). The data charting form helps to identify the concepts corresponding to the research questions. Three evaluators extracted data independently (LD, ST, MT). The form contained authors, year of publication, population, number of participants, kinesiophobic attitudes, behaviours and beliefs of health professionals, influence on interventions and patients' outcomes. The three evaluators (LD, ST, MT) agreed on each item on the data charting form. To select the suitable studies, the evaluators followed a three-step process that involved reviewing the titles, abstracts, and complete scientific articles. The data was then mapped thematically, including study descriptions, summaries of results, identification of kinesiophobic behaviour among health professionals, and the influence on both the health professional's intervention and the eventual outcomes for their patients.

# RESULTS

## Selection of studies

Overall, 2,162 potential studies have been identified throughout the databases for the review. After removing duplicates, the initial selection was made by reading the title, and then the abstracts, 43 articles remained for the entire reading. Of the forty-three, 30 articles were excluded because they did not answer the research questions where seven studies focused on the effect of intervention among health professionals, seven studies were oriented to the pain prevalence in clinicians and its impact on their job (none clinical aspects), five studies were education on treatment guidelines, three studies were conducted that specifically targeted students, three studies focused on questionnaires' validation, one was an editorial article, one article described the advice from a physiotherapist, two studies

Doutre et al. (2024), *PeerJ*, DOI 10.7717/peerj.17935

| Table 1 | Summary of characteristics of included studies. | | | | | |
|---|---|---|---|---|---|---|
| **Authors** | **Date** | **Country of publication** | **Study design and population** | **Type of patients** | **Objective of the study** | **Assessment tools** |
| *Coudeyre et al.* | 2006 | France | **Cross-sectional study** with 864 physicians | Chronic and acute lumbago | To describe the fear-avoidance beliefs of general practitioners (GPs) concerning low back pain / To investigate the impact of these beliefs on their recommendations for bed rest, physical activity and sick leave / To discover the factors associated with GPs' fear-avoidance beliefs. | Questionnaire: Fear-Avoidance Beliefs Questionnaire |
| *Daykin & Richardson* | 2004 | United Kingdom | **Qualitative study** with 6 physiotherapists and 12 patients | Chronic Low Back Pain | To study the interrelationship between the beliefs and behaviour of physiotherapists and their patients. | Semi-structured interviews Observations at designated stages |
| *Simoncsics & Stauder* | 2020 | Hungary | **Cross-sectional study** with 110 nurses, 35 physiotherapists' assistants and 30 massage therapists. | Chronic and acute lumbago | To assess clinicians' attitudes towards various treatment modalities used to treat low back pain. | Questionnaires: *Work Experience, Personal Chronic Low Back Pain, Attitudes towards therapeutic procedures*, Fear-Avoidance Beliefs Questionnaire |

**Table 1** (*continued*)

| Authors | Date | Country of publication | Study design and population | Type of patients | Objective of the study | Assessment tools |
|---|---|---|---|---|---|---|
| *Jeffrey & Foster* | 2011 | United Kingdom | **Qualitative study** using a phenomenological hermeneutical approach and practitioner-as-researcher model with 7 physiotherapists | Chronic and acute lumbago | To understand the influences on clinicians' decision-making. | Semi-structured interviews |
| *Blouin et al.* | 2019 | Canada | **Cross-sectional study** with 64 physiotherapists | Chronic Pain | To examine whether physiotherapists' knowledge of chronic pain, pain beliefs and self-efficacy to counsel on exercise predicted their intention to counsel patients with chronic pain on exercise. | Questionnaire: Health Care Providers' Pain and Disability Scale, Revised Neurophysiology of Pain Questionnaire, Self-Efficacy Scale, Intention to Prescribe Physical Activity |
| *Cross et al. (2014)* | 2013 | New Zealand | **Cross-sectional study** with 63 occupational therapists | Chronic Low Back Pain | To explore the associations between fear avoidance beliefs and occupational therapists' treatment recommendations. | Questionnaire: Fear Avoidance Belief Tool and Health Care Providers, Pains and Impairment Relationship Scale Scale: Likert Scale |
| *Linton, Vlaeyen & Ostelo (2002)* | 2002 | Sweden | **Cross-sectional study** with 60 physicians and 71 physiotherapists | Low back pain | To assess the level of fear-avoidance beliefs among clinicians | Questionnaires revised from: Tampa Scale of Kinesiophobia for Health Care Providers, Pain and Impairment Relationship Scale, Self-reported intervention behaviour |

Doutre et al. (2024), *PeerJ*, DOI 10.7717/peerj.17935

**Table 1** (*continued*)

| Authors | Date | Country of publication | Study design and population | Type of patients | Objective of the study | Assessment tools |
|---------|------|------------------------|----------------------------|------------------|------------------------|------------------|
| *Poiraudeau et al.* | 2006 | France | **Longitudinal study** on 266 Rheumatologists and 440 patients on 3 months | Chronic and acute lumbago | To assess the characteristics of clinicians influencing patient treatment. | Questionnaire: Fear-Avoidance Beliefs Questionnaire |
| *Farris et al.* | 2019 | United States | **Cross-sectional study** with 16 exercise physiologists and nurses and 117 patients | Cardiorespiratory rehabilitation | To examine the association between fear of exercise in patients and clinicians. | Questionnaire: Fear of exercise, Generalized Anxiety Disorder-7, Patient Health Questionnaire-9, Chronic Respiratory disease Questionnaire, Rand 36-item short form survey, Lung function: Forced expiratory volume in one second Exercise testing: Treadmill exercise test |
| *Fitzgerald, Hadjistavropoulos & MacNab* | 2009 | Canada | **Longitudinal study** with 85 patients and 550 professional caregivers on 3 months | Gerontology and Dementia | To investigate clinicians' fears of falls and pain for healthcare recipients. | Questionnaire: Attitudes towards falls Exercise testing: Older American Resource Services |
| *Darlow et al.* | 2011 | United Kingdom | **Systematic reviews** on 17 studies. | Chronic Low Back Pain | To investigate the association between the attitudes and beliefs of health professionals and the attitudes and beliefs, clinical management and outcomes of this patient population. | Measures of attitudes and beliefs: Fear Avoidance Beliefs Questionnaire, Pain Attitudes & Beliefs Scale, Pain & Impairment Relationship Scale, Tampa Scale for Kinesiophobia |

Doutre et al. (2024), *PeerJ*, DOI 10.7717/peerj.17935

**Table 1** (*continued*)

| Authors | Date | Country of publication | Study design and popula-tion | Type of patients | Objective of the study | Assessment tools |
|---|---|---|---|---|---|---|
| *Lakke et al.* | 2015 | Netherlands | **A blinded, cluster-randomized cross-sectional study** was performed with 256 physiothera-pists | Lifting capac-ity healthy people | To determine the influence of kinesiophobic beliefs of phys-iotherapists on lifting ability in healthy individ-uals. | Physical Capacity Test: The lifting test from Work Well protocol  Questionnaire: Tampa Scale of Kinesiophobia for Health Care Providers |
| *Gardner et al.* | 2017 | Australia | **Systematic review** on 10 studies. | Chronic Low Back Pain | To investigate an association between clini-cians' attitudes and beliefs and their clinical management. | Measures of attitudes and beliefs: Attitudes to Back Pain Scale for musculoskeletal practitioners, Back Beliefs Questionnaire adapted for paramedical therapists, Fear of Pain Questionnaire, Health Care Providers Pain and Impairment Relationship Scale, Pain Attitudes and Beliefs Scale for Physiotherapists-Biomedical orientation, Pain Attitudes and Beliefs Scale for Physiotherapists, Tampa Scale for Kinesiophobia adapted for health professionals  Measures of intervention: Moderate-risk and low-risk vignettes, Questions about level of spinal pathology, risk of developing low back pain disability, and advice to return to work and activity, Intervention questionnaire, Work-related behaviour questionnaire, Photographic Series of Daily Activities  Semi-structured interviews |
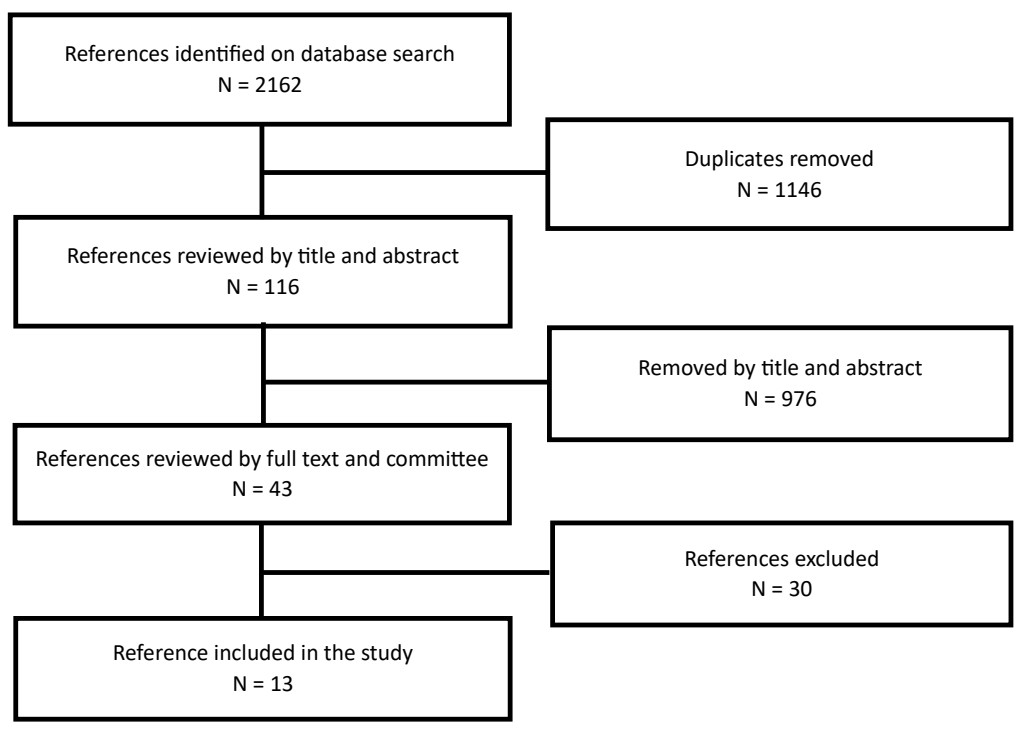

**Figure 1** **Scoping review PRISMA flow chart.**

that solely concentrate on patients and their caregivers, and one study was excluded because the publisher was identified on Beall's list (*Beall, 2023*). At the end of the process, a total of 13 articles were included in the study (Fig. 1).

## Characteristics of the studies

The thirteen studies selected explore and catalogue the extent of scientific research that identifies the causes and consequences of kinesiophobia, kinesiophobic behaviour or fear-avoidance behaviour among health professionals while they perform their intervention. Among those, 10 studies investigated the association between the professional's knowledge ($n = 2$), fear-avoidance beliefs ($n = 10$), attitudes ($n = 5$), sense of self-efficacy ($n = 1$) and fear of falling and exercising ($n = 3$) and the impact on their interventions. One study directly examines the association between the professionals' fear and those of their recipients. Two articles were systematic literature reviews. Nine articles were quantitative studies and used questionnaires. Two qualitative studies used semi-structured interviews as data collection methods. Two articles used physical capacity tests and questionnaires. Ten studies studied patients who suffered from pain, more precisely chronic pain and acute lower back pain. One article studied cardio-respiratory disorders, one studied aging people and dementia. The professionals studied were physiotherapists ($n = 6$), physicians ($n = 3$), nurses ($n = 4$), care assistants ($n = 2$), massage therapists ($n = 1$) and exercise physiologists ($n = 1$). Table 1 summarises the characteristics of the 13 included studies.

## Overview

The recent publication years of the selected articles (2002 to 2024) represent the relevance of this study and evoke the emergence of this topic. The wide variety of results provides a broad understanding of the causes and consequences of the professional's kinesiophobia in their interventions. Table 2 presents the major findings of the review divided into seven themes according to the characteristics of kinesiophobia related to professionals' intervention.

Theme 1. Kinesiophobia and kinesiophobic behaviours influence on their interventions. Two-thirds of the literature suggests that health professionals may have a wide range of kinesiophobia and kinesiophobic behaviours, which can have negative but also positive impacts on their patient's intervention. More precisely, professionals with high levels of fear-avoidance beliefs, which are related to kinesiophobia and kinesiophobic behaviours, tend to refer their patients to other health specialists less frequently, increase medication usage, and limit their patients' physical or daily activity, despite treatment guidelines recommending otherwise. Conversely, professionals with low levels of fear-avoidance beliefs tend to be more proactive by referring patients to other specialists and adhering to treatment guidelines. Seven out of eight articles were related to interventions for patients' chronic back pain, and only one article was related to geriatric patients with dementia. The health professionals assessed were mostly physiotherapists, but also physicians (general practitioners, specialists), nurses, massage therapists, occupational therapists, and professional caregivers. Theme 2. Professionals' guidance provided to their patients. Almost half of the literature identified that professionals' guidance might be influenced by their beliefs, whereas professionals tend to limit advice when they report a high level of fear-avoidance beliefs. Four out of six articles were related to the interventions with chronic back pain patients; one was for cardio-rehabilitation patients, and one was for healthy individuals. The health professionals assessed were mostly physiotherapists, but also physicians, nurses, and exercise physiologists (kinesiologists). Theme 3. Professionals' fear-avoidance beliefs. Approximately half of the literature about professionals' and patients' fear-avoidance beliefs was found to be closely related, and professionals tend to influence their intervention but non-intentionally through their beliefs. Although these six articles were all about interventions for patients with chronic back pain. The health professionals assessed were mostly physiotherapists, but also physicians (general practitioners, specialists).

Theme 4. Professionals' fear of pain. An article observed that professionals' fears of patients' falls and injuries predict the patients' loss of functional autonomy. However, the study was related to geriatric patients with dementia, and the health professionals were professional caregivers. Theme 5. Professionals' health condition and pain. Approximately one-quarter of the literature reported that professionals who experience pain were more inclined to report fear-avoidance beliefs or project themselves onto their patients or their interventions. Two articles were related to interventions with chronic back pain patients. Health professionals were physiotherapists and nurses. Theme 6. Professionals' experience. Two articles observed that the professional's level of experience influences the patient's interpretation according to their pain perception, whereas senior professionals tend to

**Table 2  Association between professionals' kinesiophobic attitudes, behaviours, beliefs, interventions and patient outcomes.**

| Themes | Outcomes discussed throughout the studies ($n = 13$) | Study |
|---|---|---|
| | - Professionals with high levels of fear-avoidance beliefs: avoid recommending treatments that will increase the physical demands; advise patients to limit work and physical activities; are more likely to recommend bed rest during sick leave; report issuing more sick notes; did not report an increase in referral in another professional. <br> - Professionals with low levels of fear avoidance beliefs are more likely to refer patients to specialists. | *Coudeyre et al. (2006)*; *Cross et al. (2014)*; *Darlow et al. (2012)*; *Poiraudeau et al. (2006)* |
| | - There is a significant association between high scores of fear-avoidance beliefs and the importance attributed to medication. | *Simoncsics & Stauder (2021)* |
| | - Professionals with biomedical orientation and high fear-avoidance beliefs were associated: with increased patient' sick leave certification or time off from work; more likely to advise patients with acute and chronic low back pain to limit work and physical activity; are less likely to adhere to treatment guidelines for low back pain. | *Gardner et al. (2017)*; *Darlow et al. (2012)* |
| Kinesiophobia and kinesiophobic behaviours influence in the professionals' intervention | - Biomedical guidance is associated with non-compliance of the directives. | *Daykin & Richardson (2004)* |
| | - Professionals with fears of falls and injury reduce patient activity. | *Fitzgerald, Hadjistavropoulos & MacNab (2009)* |
| | - Professionals with high levels of fear-avoidance beliefs provided less guidance on activities. | *Linton, Vlaeyen & Ostelo (2002)* |
| | - Professionals using biomedical orientation provided limited explanation given to patients. | *Daykin & Richardson (2004)* |
| | - Professionals with biomedical orientation and fear avoidance beliefs provide restricted advice to return to work and activities. | *Gardner et al. (2017)* |
| Professionals' guidance provided to their patients | - Professionals unintentionally maintain patients' anxiety and fear about exercise by using safety and fear-reducing behaviours. | *Lakke et al. (2015)*; *Farris et al. (2019)* |
| | - Professionals' attitudes and beliefs are associated with the type and content of guidelines provided to patients. | *Darlow et al. (2012)* |

**Table 2** (*continued*)

| Themes | Outcomes discussed throughout the studies ($n = 13$) | Study |
|---|---|---|
| Professionals' fear-avoidance beliefs | - Professionals have a range of beliefs: worrying if a patient reports pain during exercise, advising avoidance of painful movements, and believing that pain reduction is a prerequisite for returning to work. | *Linton, Vlaeyen & Ostelo (2002)* |
| | - Professional fear avoidance beliefs are associated with patient fear avoidance beliefs. | *Daykin & Richardson (2004)*; *Gardner et al. (2017)*; *Poiraudeau et al. (2006)*; *Coudeyre et al. (2006)*; *Darlow et al. (2012)* |
| Professionals' fear of pain | - Professionals' fears of falls and injuries negatively predict the patient's loss of functional autonomy. | *Fitzgerald, Hadjistavropoulos & MacNab (2009)* |
| Professionals' health conditions and pain | - Professionals with severe and moderate pain have higher fear avoidance beliefs. | *Simoncsics & Stauder (2021)*; *Lakke et al. (2015)* |
| | - Professionals who experience pain, are more inclined to project it onto their patients or their interventions. | *Daykin & Richardson (2004)* |
| | - The lack of experience or too much experience of the professionals influences the interpretation of the patient according to their painful experience. | *Daykin & Richardson (2004)* |
| Professionals' experience | - Years of experience influence the sense of self-efficacy in controlling chronic pain through exercise. | *Blouin et al. (2019)* |
| | - Advice from senior professionals influences younger colleagues. | *Daykin & Richardson (2004)* |
| Relationships with the patients | - Professionals using biomedical orientation tend to satisfy more easily their "passive" patients.<br>- In case of cognitive dissonance, professionals reduce conflict with their patients while providing a sense of satisfaction to the care, by using biomedical knowledge. | *Gardner et al. (2017)*; *Jeffrey & Foster (2012)* |

help less experienced colleagues and their patients cope with the pain. These articles were related to physiotherapists' interventions with chronic back pain patients. Theme 7. Professionals' relationships with patients. Two articles identified the relationship between health professionals and patients as a key component. Professionals tend to satisfy their "passive" patients more easily and reduce the risk of conflicts with them by avoiding dissonance. These articles were related to physiotherapists' interventions with chronic back pain patients.

## DISCUSSION

By definition, kinesiophobia reduces bodily movement and physical activity practice. Since bodily movement and physical activity are essential in multiple treatments, their influence might represent a risk of treatment failure among patients (*Huang et al., 2022*; *Perrot et al., 2018*; *Raizah et al., 2022*). To our knowledge, this article is the first scoping review to explore the impact of professionals' kinesiophobia and kinesiophobic behaviour on their patient interventions. The literature has found that the level of fear-avoidance behaviour among professionals is the most significant and widely researched variable. In addition, professionals' beliefs, knowledge, level of experience, and personal experiences with pain have also been identified as potential modulation factors. With a high level of fear-avoidance behaviour, the potential consequences could represent alterations in the patient's treatment because it tends to alienate the professional from following treatment guidelines, referring to other specialists and altering guidance provided to the patients. Consequently, the patients might risk not receiving the optimal treatment. In addition, these outcomes might contribute to an increase in the level of fear-avoidance beliefs, fear of pain, fear of exercise and physical activity of the patients. It is important to note that these potential consequences are hypotheses and require further investigation.

The current study found interesting characteristics of kinesiophobia experienced by health professionals which seem to impair their interventions as they did not follow the main advice of the fear-avoidance model (*Gardner et al., 2017*; *Coudeyre et al., 2006*; *Cross et al., 2014*; *Darlow et al., 2012*; *Daykin & Richardson, 2004*; *Fitzgerald, Hadjistavropoulos & MacNab, 2009*; *Linton, Vlaeyen & Ostelo, 2002*; *Poiraudeau et al., 2006*; *Gelso & Kline, 2019*). The characteristics of kinesiophobia from which the professional might suffer could interfere negatively with the therapeutic alliance by infiltrating (consciously or not) their beliefs, attitudes personal experiences, and health conditions. Research performed by the National Institute of Mental Health (NIMH) in the United States, shows that the therapeutic alliance has an important influence on treatment outcomes (*Scovern, 1999*). Interestingly, *Burns et al. (2008)* observe that low-quality of therapeutic alliance could lead to high levels of hostilities, expression of anger, and anxiety and depression symptoms among patients suffering from chronic pain. According to *Bordin (1979)*, the therapeutic alliance is characterized by the therapeutic tasks and goals as well as the relationship between the health professional and the patient (therapeutic bond). These characteristics continuously influence each other throughout the quality-of-life enhancement, building trust, self-esteem improvement, overall satisfaction and common beliefs between health professionals and patients (*Scovern, 1999*; *Bioy & Bachelart, 2010*). A second aspect of the therapeutic alliance is confirming the therapeutic bond (*Bordin, 1979*). The goal is to impart knowledge and skills from health professionals to patients (*Gelso & Kline, 2019*). Regrettably, in that transfer to their patients, the professionals' health status and beliefs might interfere (consciously or not) with the alliance negatively (*Lakke et al., 2015*; *Daykin & Richardson, 2004*; *Fitzgerald, Hadjistavropoulos & MacNab, 2009*; *Linton, Vlaeyen & Ostelo, 2002*; *Poiraudeau et al., 2006*; *Gelso & Kline, 2019*; *Simoncsics & Stauder, 2021*).

According to our findings, the health professional's health status and fear of pain seem to interfere with the therapeutic bond.

The therapeutic bond represents how the patient and the professional feel understood, respected and valued, particularly patients whom professionals identify as "passive" (*Bordin, 1979*). Patients expected more from their health professionals and asked, rightfully, for higher standards of care and service (*Downey-Ennis & Harrington, 2002*). Patient satisfaction has become a frequent outcome of the quality of care and service delivery. In that sense, satisfaction represents a positive appraisal of provided treatment concerning the patient's goals and expectations (*Lochman, 1983*). However, to satisfy the patient, the professional may perform a biomedical and also, a more "kinesiophobia-prevention" approach (or following the advice from the fear-avoidance model), to avoid conflict and cognitive dissonance with the patient (*Jeffrey & Foster, 2012*).

### Limitations

It is important to understand that this scoping review, while comprehensive, is not a systematic review or quantitative study. Therefore, the results should not be generalized or interpreted as cause–effect relationships. While extensive, the use of three databases (Medline, SportDiscus, and CINAHL) in this study may limit the breadth of our findings and interpretations of the results. In addition, it's important to mention that a large portion of the literature we found focused on interventions by physiotherapists, with the most frequently studied health condition being chronic back pain. This limits also the applicability of the findings to other healthcare professionals and health conditions, highlighting the need for future research to explore a wider range of kinesiophobia manifestations. No selected studies directly address kinesiophobia or kinesiophobia behaviours among health professionals but studied fear-avoidance beliefs among health professionals, which highlights the need for further research to address directly this topic.

### Practical implications

Even though the generalization is difficult, it appears that professionals' kinesiophobia and kinesiophobic symptoms are important issues that lead to negative outcomes among patients. Health professionals should be more aware of the impacts of their kinesiophobic beliefs and behaviours in their interventions. Health professionals can benefit from training, debriefing and skill building in physical activity, and how to manage to include physical activity in their interventions should identify the psychosocial needs of the patients and adopt an interdisciplinary/multidisciplinary approach to ease referrals and comprehensive, coordinated care of patients (*Gardner et al., 2017*; *Fitzgerald, Hadjistavropoulos & MacNab, 2009*). Health professionals should empower their patients to understand their disease, encourage self-reliance, and promote patient participation.

## CONCLUSION

In recent years, the importance of physical activity practice has gained widespread recognition, not only for promoting overall health but also as a therapeutic approach for various health conditions. However, health professionals often encounter patients who

harbour a fear of bodily movement and avoid physical activity altogether. Compounding this challenge, our study reveals that health professionals themselves may experience kinesiophobia or exhibit kinesiophobic behaviour, potentially impacting their interventions and ultimately, patient outcomes. Our research aimed to document the outcomes of these kinesiophobic attitudes, behaviours, and beliefs among professionals, uncovering notable associations with their fear-avoidance behaviours, but also professional experience and knowledge, fear of pain, and health status or pain. These findings suggest potential ramifications for patient behaviours and outcomes, underscoring the critical need for further investigation. Moreover, health professionals must recognize the potential detrimental effects of kinesiophobia on patient care. The insights gleaned from our study can inform the development of cognitive and behavioural interventions tailored for health professionals aimed at enhancing the utilization of physical activity prescriptions for patients with chronic conditions and traumatic injuries. Moving forward, there is a clear imperative for additional research on the concept of professional kinesiophobia to bolster the quality of care for patients and enhance the efficiency of the healthcare system. Future studies should delve deeper into kinesiophobia in health professionals among diverse professionals and its correlation with therapeutic adherence to physical activity recommendations. By addressing these gaps, we can advance our understanding and implementation of strategies to mitigate kinesiophobia, ultimately improving patient care and outcomes in clinical practice.

### Funding
The authors received no funding for this work.

### Competing Interests
The authors declare there are no competing interests.

### Author Contributions
- Lilian Doutre conceived and designed the experiments, performed the experiments, analyzed the data, prepared figures and/or tables, authored or reviewed drafts of the article, and approved the final draft.
- Maryse Beaumier conceived and designed the experiments, authored or reviewed drafts of the article, and approved the final draft.
- Andrée-Anne Parent conceived and designed the experiments, authored or reviewed drafts of the article, and approved the final draft.
- Sébastien Talbot performed the experiments, authored or reviewed drafts of the article, and approved the final draft.
- Mathieu Tremblay conceived and designed the experiments, performed the experiments, authored or reviewed drafts of the article, and approved the final draft.

### Data Availability
This is a literature review and no datasets were generated during the current study.

## Supplemental Information

Supplemental information for this article can be found online at http://dx.doi.org/10.7717/peerj.17935#supplemental-information.

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
