# Peer review of "Kinesiophobia among health professionals’ interventions: a scoping review"

_PeerJ, doi:10.7717/peerj.17935_

## Round 0.1 · original submission · Major Revisions

· Academic Editor

Major Revisions

Dear authors.

Please read the reviewers' comments carefully and make changes appropriately.

In case you decide not to proceed with a change, please justify adequately the reason for this decision.

Reviewer 1 ·

Basic reporting

Clear and unambiguous, professional English used throughout.
I do not have a native level of English to state if use of English is appropiate. However, there are strange aspects, such as in the table of results citing the authors with "and al" instead of "et al".

Literature references, sufficient field background/context provided.
In the manuscript the authors use a sufficient number of references to justify the work.

Professional article structure, figures, tables. Raw data shared.
This review covers the clinically important target of the presence of aspects of kinesiophobia and fear of movement within healthcare professionals.

The study is a scoping review and this is where my concerns lie. There is no qualitative analysis of the evidence, because it is not a Systematic Review, nor include quantitative analysis, because it is not a Meta-analysis. However, although the study design justifies not carrying out these analyses, I am concerned about their applicability and subjective interpretation of the results.

Moreover, in the Introduction section, the authors provide information that should be in the Methods section "The scoping review has been conducted on the CINHAL, Pubmed, and Sportdiscus databases as of November 2022".

Is the review of broad and cross-disciplinary interest and within the scope of the journal?
Yes, the paper covers all types of health professionals dealing with pain patients.

Has the field been reviewed recently? If so, is there a good reason for this review (different point of view, accessible to a different audience, etc.)?
The authors provide the date (November 2022) on which the search was conducted, so I consider the search to be too old, and therefore, it should be updated.

In addition, we can see that the review itself includes 2 systematic reviews (Darlow et al.2015 and Gardner et al 2017) that address Fear Avoidance Beliefs, Pain Attitudes & Beliefs, Kinesiophobia in clinicians that handled chronic and non-chronic low back pain patients. In this scoping review 9 out of 13 included studies are related to low back pain. Therefore, given that two well-structured systematic reviews have already been published on this topic, my concerns is what new aspects this study brings to the literature and also to clinicians because the justification of “no scoping review has been found on this topic” does not justify it. Although the approach in this review is broader and covers more ailments than low back pain, I do not see the point of including patients with low back pain.

On the other hand, I am also concerned that for the purpose of this review “to explore and catalogue the extent of scientific research that identifies the causes and consequences of kinesiophobia, kinesiophobic behaviour or fear-avoidance behaviour among health professionals while they perform their interventions”, cross-sectional studies have included in the review, where no cause-effect relationship can be extracted from them.

Does the Introduction adequately introduce the subject and make it clear who the audience is/what the motivation is?
Yes it does

Experimental design

Article content is within the Aims and Scope of the journal.
Yes

Rigorous investigation performed to a high technical & ethical standard.
Within the review designs, the scoping review presents a less rigorous design than the Systematic Reviews and Meta-analysis as they do not present an analysis of the Methodological Quality nor do they carry out a Qualitative and/or Quantitative Analysis of the studies included. Thus, the study design itself implies less rigorous research than other designs such as the two Systematic Reviews already published on the same topic. This aspect concerns me as I do not know what this research contributes to what has been published to date. It would have been interesting to update the last systematic review carried out.

Methods described with sufficient detail & information to replicate.
PubMed is not a database (Medline)
Author state “To select the suitable studies, the evaluators followed a three-step process that involved reviewing the titles, abstracts, and complete scientific articles”. However, in figure 1, steps 1 and 2 are merged as only one

Are sources adequately cited? Quoted or paraphrased as appropriate?
PeerJ uses the style 'Name. Year' with an alphabetical list of references. Authors use numbers in brackets in the text and the reference list is sorted in order of appearance rather than alphabetically.

Is the review organized logically into coherent paragraphs/subsections?
Yes It is

Validity of the findings

Conclusions are well stated, linked to original research question & limited to supporting results.
The review includes several cross-sectional studies, from the cross-sectional studies no conclusions can be drawn, such as “To maintain a positive therapeutic bond (alliance), health professionals must be mindful of their fear of movement and how it can impact treatment outcomes”

Is there a well developed and supported argument that meets the goals set out in the Introduction?
Given that two well-structured systematic reviews have already been published on this topic, my concerns is what new aspects this study brings to the literature and also to clinicians because the justification of “no scoping review has been found on this topic” does not justify the goal.

Does the Conclusion identify unresolved questions / gaps / future directions?
My concern is the interpretation of the results of the review. The conclusion meets its objective, however, by interpreting results from cross-sectional observational studies we cannot establish some parts of the conclusion where it is mentioned that monitoring kinesiophobia in health professionals may improve the use of physical activity prescription for the management of patients with chronic pain. Perhaps future studies should analyse the effectiveness of managing kinesiophobia in health professionals and its relationship to therapeutic adherence to physical activity.

Additional comments

If the authors are willing to make these changes (do a systematic review and an objective interpretation of the results) it will be considered for a future publication. If these changes are not made, I do not see that the study has the necessary quality to be published in this journal.

·

Basic reporting

The manuscript presents a relevant topic within the field, aiming to catalog research on kinesiophobia among health professionals. While the subject is well-chosen and the structure of the article initially appears to meet professional standards, the abstract requires refinement. The background information, though informative, occupies too much of the abstract, overshadowing the study's conclusions. It is advised that the authors provide a succinct background and a more comprehensive summary of the study's findings and implications.Regarding the introduction, a reference is requested to support the statement on kinesiophobia being studied as a measure of intervention efficiency. The literature review, while methodologically based on established guidelines, raises the question of why only three databases were utilized. Further justification for this selection is necessary to ensure comprehensive coverage of the topic.

Experimental design

no comment

Validity of the findings

The conclusions touch upon the association between findings and patient behaviors and outcomes. If direct measurement of this impact is not within the scope of the review, it should be presented as a potential avenue for future research rather than a current finding.

Additional comments

This manuscript entitled “Kinesiophobia among health professionals: interventions: a scoping review,” primarily aimed to catalog the scientific research on the causes and consequences of kinesiophobia in this demographic, is commendable, as it may have implications for patient care and the well-being of health professionals themselves. While it is a very interesting topic, But I think this manuscript has some flaws to fill in before it can be published in a journal. There are several questions that should be addressed, which are listed below.
1. In the abstract part, In the opinion of reviewer, the author provided too much background descriptions in this part, which may be too long-winded. I suggest that the authors provide more detailed descriptions of the conclusions of this study in this part.
2. “The literature review was based on the Joanna Briggs Institute manual and the PRISMA method for a scoping review. The research has been conducted on three databases.” Why were only 3 databases selected by the authors.
3. In the Introduction part, “Kinesiophobia has been studied among patients and continues to be studied as a measure of intervention efficiency.” Please provide a reference to support this sentence.
4. Begin with a broader context of kinesiophobia, establishing its relevance in health sciences. Then, narrow down to the specific issue of kinesiophobia among health professionals, highlighting why this is an important and under-researched area.
5. "A positive relationship is observed between kinesiophobia symptoms and the likelihood of returning to sports after surgery . " The term "positive relationship" is ambiguous and could be misinterpreted as a beneficial correlation. Clarify this statement to reflect the nature of the relationship accurately. For instance: "Increased symptoms of kinesiophobia are associated with a decreased likelihood of returning to sports after surgery."
6. In the results part, "The fear-avoidance model helps to understand the association between pain-related fear and safety-seeking behaviours ." The introduction would benefit from a brief explanation of how the fear-avoidance model applies to health professionals. Is there evidence that this model is valid for this group, or is the application hypothetical? This clarification will set the stage for why the model is relevant to the review.
7. In the discussion part, "The results showed that the professionals' kinesiophobia behaviour had a negative impact on their interventions and thus on their patients." Clarify the nature of the impact by specifying whether the studies demonstrated a direct causal link or if the negative impact is inferred based on observed associations. Also, detail the types of negative impacts observed and how they were measured.
8. "The link between kinesiophobia and the therapeutic alliance is natural through the fear-avoidance model." Explain why the link is considered 'natural' and discuss whether the studies explicitly examined the therapeutic alliance in the context of kinesiophobia, or if this is a theoretical extrapolation.
9. "The consequences were alterations in the patient’s treatment because it tends to alienate the professional from following treatment guidelines, referring to other specialists and altering guidance provided to the patients." Provide evidence from the studies that supports the claim that kinesiophobia leads to alterations in patient treatment. If the evidence is not direct, it would be appropriate to indicate that these are potential consequences that need further investigation.
10. In the conclusion part, "These findings are also associated with their patients' behaviours and outcomes." Provide clarity on the nature of the association. If the study that you included did not directly measure the impact on patients' behaviours and outcomes, it should be stated that these are potential associations that require further research.

---

## Round 0.2 · Major Revisions

· Academic Editor

Major Revisions

After reviewing the article and the review received, it is considered that the paper requires important changes to be considered for publication in PeerJ.

Please respond appropriately and with justification to the reviewer's considerations.

·

Basic reporting

Please find detailed comments below.

Experimental design

Please find detailed comments below.

Validity of the findings

no comment.

Additional comments

In this study, the authors sought to explore and catalogue the extent of scientific research that identifies the causes and consequences of kinesiophobia among health professionals while they perform their interventions. The purposes of this study appear to be specific, but the research novelty and results should be further highlighted and improved.
1. Abstract, background, the background information could be improved to better emphasize the novelty and significance of the research.
2. Abstract, the methods section should briefly introduce what search terms were used and which databases were searched.
3. Abstract, the results section of a scoping review should go beyond just listing the identified research themes or topics. It is recommended that the authors include a concise narrative that describes the overall state of the research in the area. By incorporating this broader narrative summary, the results section can give readers a more comprehensive understanding of the current knowledge and landscape within the field of study.
4. The conclusions are so broad, please rewrite it based on the findings of your study. Meanwhile, it would be beneficial to improve the quality of the keywords used to make it easier for researchers and other interested parties to locate information relevant to the research topic.
5. Introduction, the author introduced the extent of scientific research that identifies the causes and consequences of kinesiophobia among health professionals while they perform their interventions. I could not see the gap that the authors were going to bridge, and neither can I see the significance of this study properly. What is the current research status in this field? Are there any similar or related review studies? What are the differences or advantages of this article compared to the previous research? Why is it important to conduct such research on health professionals? These questions should be clearly explained in this part. “Since this topic is still understood yet.”, if the research in this area is limited, then it would be logical to suggest that more experimental or empirical research studies are needed, rather than just more scoping or narrative reviews.
6. Methods, “2.1 Identifying the research questions”, in the introduction section, the authors should elaborate in greater detail on the specific research questions being addressed. They should clearly explain the importance and rationale behind investigating these particular research questions, highlighting their significance and relevance within the broader context of the field.
7. “The scoping review has been conducted on the CINHAL, Pubmed, and Sportdiscus databases as of January 2024.” Please update the searching.
8. In the results section of this scoping review, the authors should thoroughly describe the current state of research for each of the key topical areas or themes that were identified. This detailed narrative on the research status of each domain should mirror the summary provided in the abstract. Within the results, the authors should not only list the main research topics, but also delve into the specifics of what is known and not known in each area. This should include outlining the limitations and gaps in the existing literature, as well as indicating potential avenues for future investigation and research. By providing this comprehensive overview of the research landscape for each key area in the results section, the authors can give readers a more complete picture of the current knowledge and outstanding questions in the field. This level of detail can strengthen the scoping review's ability to map the existing research and identify priority areas for future study.
9. “The most important limitation is that a high proportion (nearly 77 %) of the article was related to chronic back pain, which limits the generalization to other health conditions or injuries.”, These are not the limitations of the review, but the limitations of the preliminary research.
10. The conclusions should be further strengthened based on the main findings of this study.

---

## Round 0.3 · accepted · Accept

· Academic Editor

Accept

I confirm that the authors have addressed all of the reviewers' comments.
Now, the peper is ready to be published.
Thank you.

·

Basic reporting

no comment.

Experimental design

no comment.

Validity of the findings

no comment.

Additional comments

The authors have addressed all previous comments, I suggested to be published without further modification.